# Analysis of the Sensitivity and Specificity of Histopathological Findings for Diagnosing Lupus Nephritis

**DOI:** 10.3390/diagnostics14232681

**Published:** 2024-11-27

**Authors:** Epitácio Rafael da Luz Neto, Maria Brandão Tavares, Ana Gabriela de Jesus Torres de Melo, Washington L. C. dos-Santos, Denise Maria Avancini Costa Malheiros, Luís Yu

**Affiliations:** 1Division of Nephrology, Hospital das Clínicas, University of São Paulo School of Medicine, São Paulo 05403-010, SP, Brazil; 2Hospital Ana Nery, Salvador 40320-010, BA, Brazil; 3Division of Nephrology, Hospital Universitário Professor Edgar Santos, Federal University of Bahia, Salvador 40110-060, BA, Brazil; 4Gonçalo Moniz Institute, Oswaldo Cruz Foundation, Salvador 40296-710, BA, Brazil; 5Department of Pathology, University of São Paulo School of Medicine, São Paulo 05403-000, SP, Brazil

**Keywords:** lupus nephritis, systemic lupus erythematosus, renal biopsy, histopathologic features, glomerular diseases

## Abstract

Background: Since the introduction of the SLICC criteria in 2012, biopsy-proven lupus nephritis (LN) has been the only independent diagnostic criterion for systemic lupus erythematosus (SLE). This was reaffirmed by the EULAR/ACR in 2019, emphasizing the importance of renal biopsy in LN. However, the current classification lacks specific histopathological criteria for defining LN. This study describes the histological findings of patients with LN, compares them with those of other glomerular diseases, and evaluates their diagnostic accuracy in a large Latin American population. Methods: This retrospective cohort included 731 kidney biopsies from two distinct academic centers. The patients were divided into two groups as follows: a LN group and a control group comprising patients with membranous nephropathy, IgA nephropathy, membranoproliferative glomerulonephritis, pauci-immune glomerulonephritis, and proliferative glomerulonephritis. Sensitivity and specificity analyses were conducted for various histopathological features. Results: We identified the following five features strongly correlated with LN: mesangial proliferation, subendothelial deposits, C1q staining ≥1+, dominant IgG, and ≥4 positive immunofluorescence elements. Combined, these features yielded an area under the ROC curve of 0.94 (95% CI: 0.91–0.95). These results were validated in a diverse population. In membranous nephropathy, histological features such as mesangial deposits, C1q positivity, and ≥4 positive immunofluorescence elements effectively distinguished class V LN from non-lupus membranous nephropathy, with an area under the ROC curve of 0.85 (95% CI: 0.76–0.93). Conclusions: The combination of mesangial proliferation, subendothelial deposits, C1q staining ≥1+, dominant IgG, and ≥4 positive immunofluorescence elements offer good accuracy for diagnosing renal involvement in SLE in a large Latin American population. In the absence of pathognomonic features, combined criteria are valuable diagnostic tools, particularly when other SLE criteria are lacking.

## 1. Introduction

Lupus nephritis (LN) is one of the most severe manifestations of systemic lupus erythematosus (SLE) [1,2,3]. It is considered the leading secondary cause of glomerular diseases, excluding diabetic nephropathy [4,5,6,7]. SLE behavior differs across different populations according to ethnicity and geographical distribution, and renal involvement occurs more frequently and with worse outcomes in non-Caucasian populations, such as the Latin American population, resulting in a higher progression to end-stage renal disease (ESRD) [2,8]. More than 25% of patients with severe LN are expected to progress to ESRD after 20 years [1,3,9].

Several updates have been made since the first World Health Organization (WHO) LN classification in 1974, the most recent being the update by the International Society of Nephrology/Renal Pathology Society (ISN/RPS) in 2003 [10,11]. Nonetheless, renal biopsy cannot be used in isolation for the diagnosis of SLE; it must be interpreted within the appropriate clinical context, necessarily, including serological markers.

Formerly, LN was diagnosed by demonstrating proteinuria > 500 mg/day or 3+ and abnormal cylindruria in a patient meeting at least four out of the eleven American College of Rheumatology (ACR) criteria [12,13]. In 2012, the Systemic Lupus International Collaboration Clinics (SLICC) established that biopsy-confirmed renal involvement alone was definitive for SLE diagnosis without needing additional criteria [14]. In 2019, the ACR and the European League Against Rheumatism (EULAR) introduced a new classification confirming the renal criterion, albeit with some differences from SLICC. A diagnosis of LN class II or V contributes 8 points and class III or IV contributes 10 points, which is sufficient to diagnose SLE (minimal 10 points) [15,16], along with one serological biomarker. In this situation, especially in the absence of other lupus clinical and serological manifestations, a diagnosis of SLE is established, and consequently, treatment is initiated.

Currently, none of the lupus nephritis histopathological findings alone are sufficiently specific to definitively ascertain LN. However, certain histological findings strongly suggest LN. For instance, full-house staining is one of the most common histopathological characteristics associated with LN, as well as strong immunofluorescence C1q staining. Other LN-suggestive characteristics include tissue antinuclear factors in immunofluorescence, tubuloreticular inclusions in electron microscopy, extraglomerular deposits in immunofluorescence and/or electron microscopy, and combined subepithelial and subendothelial deposits [17,18,19]. Therefore, it remains undecided which histopathological alterations are most specific for LN diagnosis.

This study aimed to describe the histological findings in LN biopsies and compare them with those of other glomerular diseases presenting similar clinical features, using a large database from two distinct Latin American academic centers. The objective of this study was to identify the histopathological findings most specific to LN in an ethnically diverse population.

## 2. Materials and Methods

### 2.1. Population and Sampling

This retrospective, observational cohort study focused on individuals aged >14 years who underwent renal biopsy at two major nephrology referral centers of Brazil’s public health system, both associated with public universities, as follows: Hospital Ana Nery (HAN), affiliated with the Federal University of Bahia in Salvador, Bahia; and Hospital das Clínicas of the University of São Paulo Medical School (HCFMUSP) in São Paulo, São Paulo.

All individuals who underwent renal biopsy between January 2015 and December 2019 at HCFMUSP and between January 2014 and December 2020 at HAN with a reported diagnosis of LN, IgA nephropathy, membranous nephropathy, pauci-immune glomerulonephritis, membranoproliferative glomerulonephritis, crescentic and/or necrotizing glomerulonephritis, or proliferative glomerulonephritis (mesangial, focal, diffuse, or unspecified) were included in the study. All these diseases were included because of their similar histological alterations, making them amenable for comparison. The exclusion criteria were renal biopsies with insufficient material, inconclusive results, or diagnoses of diabetic nephropathy, thrombotic microangiopathy, minimal change disease, focal segmental glomerulosclerosis, isolated tubulointerstitial nephropathy, and hereditary nephropathy. These latter diagnoses were excluded because usually these diseases have a typical clinical presentation which differs from LN clinical manifestations, and their histopathological findings are distinctive from LN. If a patient had more than one biopsy, the second was excluded unless the first was inconclusive or lacked sufficient material, in which case only the second biopsy was included. Patients were classified as LN if they met the SLICC 2012 and/or ACR/EULAR 2019 criteria.

### 2.2. Data Assessment

Renal biopsies were indicated by the attending nephrologists according to the commonly recommended indications and personal experience. For HAN, the fragments for optical microscopy were placed in an acidic alcoholic formalin container. At HCFMUSP, some specimens were stored in Duboscq-Brasil solution, while others were stored in formalin. Besides this sample storage difference in some biopsies, all the other procedures were similar between the 2 institutions, including the staining methods. Fragments for immunofluorescence microscopy were transported in 0.9% sodium chloride solution at 4–8 °C, embedded in Tissue-Tek O.C.T. compound, frozen in liquid nitrogen, sectioned in a cryostat, and incubated with antibodies against IgA, IgG, IgM, kappa, lambda, C3, C1q, and fibrinogen. Electron microscopy is not routinely performed by either service, which precluded analysis of some EM histological findings, such as tubuloreticular inclusions, extraglomerular deposits, and combined subepithelial/subendothelial deposits. All HAN samples were analyzed by the same pathologist at the Gonçalo Moniz Research Centre of Fiocruz-BA, and those from HCFMUSP were analyzed by a team of pathologists from the Pathology Division.

The principal investigator collected data from reports provided by pathology services and medical records at both research sites using a data collection instrument. The following histological variables were analyzed: histopathological diagnosis; mesangial proliferation; endocapillary hypercellularity; cellular, fibrocellular, and fibrous crescents; mesangial, subendothelial, subepithelial, tubular, vascular, and extraglomerular deposits; immunofluorescence positivity and intensity for IgA, IgG, IgM, C3, C1q, kappa, lambda, and fibrinogen with any result from trace (±) onwards considered positive; global glomerulosclerosis, tubular atrophy, and interstitial fibrosis; and acute and chronic vascular changes. The immunofluorescence pattern was considered ‘full house’ when positivity for IgA, IgG, IgM, C3, and C1q was observed.

### 2.3. Statistical Analysis

The patients were divided into the reference LN group and a control group of non-SLE cases. Qualitative variables are expressed as absolute frequencies and proportions. Numerical variables not normally distributed were summarized as medians and interquartile ranges. Missing or incomplete data were excluded from the analysis.

Logistic regression analyses were performed to identify the histopathological characteristics associated with LN. The sensitivity, specificity, accuracy, positive and negative predictive values, and positive and negative likelihood ratios were analyzed for each correlated factor. The same tests were conducted to assess the accuracy of differentiating class V LN from non-lupus membranous nephropathy. Multivariate analysis was performed to identify features independently correlated with LN, followed by multinomial logistic regression to determine the best combination of histopathological elements with high accuracy. These analyses were used to construct the receiver-operating characteristic (ROC) curves. All analyses were conducted using R software (version 4.2.2) with 95% confidence intervals (CIs).

The researchers followed local ethics guidelines for human research. The project was approved by the ethics committees of Hospital Ana Nery in Salvador and Hospital das Clinicas at the University of São Paulo Medical School. As this was a retrospective observational study, an exemption from obtaining informed consent was requested and granted. The researchers also signed a confidentiality agreement for the information obtained from medical records and renal biopsy reports.

## 3. Results

A total of 731 patients were included, 488 from HCFMUSP and 243 from HAN. The LN group comprised 433 patients, 269 from HCFMUSP and 164 from HAN. The control group comprised 298 individuals, 219 from HCFMUSP and 79 from HAN.

### 3.1. HCFMUSP Population

#### 3.1.1. General Characteristics

Patients with LN at HCFMUSP were younger (median age 31.5 years) and predominantly female (84.8%). Most patients were white (68%), with a significant proportion of black and mixed-race individuals (24.9%). The control group was older (median age, 44 years) and included more men (55.2%). Patients in the LN group had better renal function, with a median baseline creatinine level of 1.00 mg/dL compared to 1.96 mg/dL in the control group. Proteinuria and hemoglobin, albumin, and cholesterol levels differed slightly between the groups (Table 1).

#### 3.1.2. Histological Diagnosis

The most prevalent histological class of LN was the diffuse form (class IV), accounting for 38.7% of the histopathological reports, followed by the focal (class III) and membranous (class V) forms, both accounting for 17.1% of cases (Appendix A). In the control group, IgA nephropathy was the most prevalent diagnosis (40.2%), followed by membranous nephropathy (21.5%), pauci-immune glomerulonephritis (14.6%), membranoproliferative glomerulonephritis (including C3 glomerulopathy) (14.3%), and proliferative glomerulonephritis (9.6%) (Appendix A).

#### 3.1.3. Optical Microscopy—HCFMUSP

Mesangial proliferation was found in 88.8% of the LN biopsies and in 63.9% of the control group. Endocapillary hypercellularity was present in 64% and 36.5% of patients in the LN and control groups, respectively. Cellular and/or fibrocellular crescents appeared in 53.8% of the LN biopsies and 33.8% of the controls. Interstitial fibrosis and tubular atrophy (IF/TA) percentages were similar to global glomerulosclerosis among the groups. A high degree of chronicity, defined as global glomerulosclerosis and/or IF/TA > 50%, was observed in 11.4% of LN biopsies, and 18.1% of the control group. Acute vascular changes were noted in 14.5% of patients with LN and 12.8% of the controls, while chronic vascular changes occurred in 67.7% and 74.4% of the patients, respectively (Appendix A).

#### 3.1.4. Characterization of Immune Deposits—HCFMUSP

Mesangial, subendothelial, and subepithelial deposits were present in 87.3%, 73.8%, and 49.2% of LN biopsies and 77.8%, 24.2%, and 23.5% of the control group, respectively. Combined subepithelial and subendothelial deposits occurred in 29.8% of the LN cases and only 1.8% of the controls. Extraglomerular deposits were present in 15.7% and 2.8% of the LN and control groups, respectively. Tubulointerstitial deposits were scarce, but they were more frequent in the lupus nephritis group (Table 2).

In LN, IgG was positive in 97.6% of cases, followed by C3 (91.5%). In the control group, C3 was the most prevalent (75.5%), followed by lambda (64.9%), and IgA (49.0%). Full-house staining was found in 36.8% of the LN cases and in only 3.8% of the control group. By removing the requirement for IgA or IgM, prevalence increased to 71.3% in the LN group and 6.2% in the control group. When considering the positivity of at least four of the five immunofluorescence elements indiscriminately, the prevalence in the LN group increased to 75.7% and to 10.7% in the control group. LN showed dominant IgG staining in 91.5% of patients compared to 32.5% in the control group. C1q staining ≥1+ was found in 79.4% of LN and 11.1% in the control group.

#### 3.1.5. Histopathological Features Diagnostic Performance—HCFMUSP

Table 3 presents the sensitivity and specificity of histopathological features. Mesangial proliferation had 89% sensitivity and 36% specificity and endocapillary hypercellularity had 64% and 63% for lupus nephritis. The mesangial area was the most frequent deposition site in lupus nephritis, with 87% sensitivity, followed by the subendothelial area, with 74% sensitivity and 76% specificity. Extraglomerular and combined subendothelial and subepithelial deposits showed very high specificity (97% and 98%, respectively) but low sensitivity. The immunofluorescence elements that best correlated with lupus nephritis were IgG and C1q. Although IgG showed the highest sensitivity (98%), C1q showed the highest specificity (82%). IgG staining as a dominant or co-dominant element improved the specificity from 60% to 67%. C1q ≥1+ increased its specificity from 82% to 89% and reduced its sensitivity from 87% to 79%. The full-house pattern showed 37% sensitivity and 96% specificity. When considering the positivity of the four elements with IgG, C3, and C1q as requirements and IgA or IgM as a fourth positive element, the sensitivity was 71% and specificity was 94%. However, the most balanced performance was obtained when considering positivity for the four elements indiscriminately, with 76% sensitivity and 89% specificity.

#### 3.1.6. LN Histopathological Criteria Combination—HCFMUSP

Mesangial proliferation, subendothelial deposits, C1q positivity ≥1+, dominant or co-dominant IgG, and four or more positive elements in immunofluorescence remained independently related to the diagnosis of LN in the multivariate analysis. Table 4 shows the sensitivity and specificity values for one or more combined histopathological elements. The best performance was found for a combination of three or more factors, with 91% sensitivity and 87% specificity, and four or more factors, with 74% sensitivity and 96% specificity. The area under the ROC curve for these five histopathological features in distinguishing LN from non-lupus nephropathy was 94%. (Figure 1).

**Table 3 diagnostics-14-02681-t003:** Diagnostic performance of histopathological features for lupus nephritis–HCFMUSP.

Feature	Sensitivity (IC 95%)	Specificity (IC 95%)	Accuracy (IC 95%)	PPV (IC 95%)	NPV (IC 95%)	Positive LR (IC 95%)	Negative LR (IC 95%)
Mesangial proliferation	0.89 (0.84, 0.92)	0.36 (0.30, 0.43)	0.65 (0.61, 0.69)	0.63 (0.58, 0.68)	0.72 (0.63, 0.81)	1.39 (1.25, 1.55)	0.31 (0.21, 0.46)
Endocapillary hypercellularity	0.64 (0.58, 0.70)	0.63 (0.57, 0.70)	0.64 (0.59, 0.68)	0.68 (0.62, 0.74)	0.59 (0.53, 0.65)	1.75 (1.44, 2.13)	0.57 (0.47, 0.68)
Mesangial deposits	0.87 (0.83, 0.91)	0.22 (0.17, 0.28)	0.58 (0.53, 0.62)	0.57 (0.52, 0.62)	0.59 (0.48, 0.70)	1.12 (1.03, 1.22)	1.12 (1.03, 1.22)
Subendothelial deposits	0.74 (0.68, 0.79)	0.76 (0.70, 0.81)	0.75 (0.71, 0.79)	0.79 (0.73, 0.84)	0.70 (0.64, 0.76)	3.05 (2.38, 3.91)	0.35 (0.28, 0.43)
Subepithelial deposits	0.49 (0.43, 0.55)	0.76 (0.70, 0.82)	0.62 (0.57, 0.66)	0.72 (0.65, 0.78)	0.55 (0.50, 0.61)	2.10 (1.60, 2.74)	0.66 (0.58, 0.76)
Combined subendothelial and subepithelial deposits	0.30 (0.24, 0.36)	0.98 (0.95, 0.99)	0.61 (0.56, 0.65)	0.95 (0.88, 0.99)	0.54 (0.48, 0.59)	16.08 (5.98, 43.20)	0.72 (0.66, 0.78)
Extraglomerular deposits	0.16 (0.11, 0.21)	0.97 (0.94, 0.99)	0.53 (0.48, 0.58)	0.87 (0.73, 0.95)	0.49 (0.44, 0.54)	5.51 (2.38, 12.75)	0.87 (0.82, 0.92)
Positive IgA	0.57 (0.50, 0.63)	0.51 (0.44, 0.58)	0.54 (0.49, 0.59)	0.58 (0.52, 0.64)	0.50 (0.43, 0.57)	1.16 (0.97, 1.38)	0.85 (0.70, 1.03)
Positive IgG	0.98 (0.95, 0.99)	0.60 (0.53, 0.66)	0.80 (0.76, 0.84)	0.74 (0.69, 0.79)	0.95 (0.90, 0.98)	2.42 (2.05, 2.86)	0.04 (0.02, 0.09)
Positive IgM	0.70 (0.63, 0.75)	0.57 (0.50, 0.64)	0.64 (0.59, 0.68)	0.66 (0.60, 0.72)	0.61 (0.54, 0.68)	1.61 (1.35, 1.92)	0.53 (0.43, 0.67)
Positive C3	0.92 (0.87, 0.95)	0.25 (0.19, 0.31)	0.61 (0.56, 0.65)	0.59 (0.54, 0.64)	0.71 (0.59, 0.81)	1.21 (1.11, 1.32)	0.35 (0.22, 0.55)
Positive C1q	0.87 (0.82, 0.91)	0.82 (0.76, 0.87)	0.84 (0.81, 0.88)	0.85 (0.80, 0.89)	0.84 (0.78, 0.88)	4.72 (3.53, 6.32)	0.16 (0.12, 0.23)
C1q ≥1+	0.79 (0.74, 0.84)	0.89 (0.84, 0.93)	0.84 (0.80, 0.87)	0.90 (0.85, 0.93)	0.78 (0.72, 0.83)	7.15 (4.84, 10.56)	0.23 (0.18, 0.30)
Dominant IgG	0.92 (0.87, 0.95)	0.67 (0.61, 0.74)	0.81 (0.77, 0.84)	0.77 (0.72, 0.82)	0.87 (0.80, 0.92)	2.82 (2.30, 3.44)	0.13 (0.08, 0.19)
Full-house staining	0.37 (0.31, 0.44)	0.96 (0.93, 0.98)	0.64 (0.60, 0.69)	0.92 (0.85, 0.96)	0.56 (0.51, 0.62)	9.73 (4.84, 19.57)	0.65 (0.59, 0.72)
IgG/C3/C1q plus IgA or IgM	0.71 (0.65, 0.77)	0.94 (0.90, 0.97)	0.82 (0.78, 0.85)	0.93 (0.89, 0.96)	0.73 (0.68, 0.79)	11.46 (6.73, 19.51)	0.31 (0.25, 0.37)
≥4 elements staining in IF	0.76 (0.70, 0.81)	0.89 (0.84, 0.93)	0.82 (0.78, 0.85)	0.89 (0.84, 0.93)	0.75 (0.70, 0.81)	7.09 (4.75, 10.59)	0.27 (0.22, 0.34)

HCFMUSP, Hospital das Clinicas of the University of São Paulo Medical School; PPV, positive predictive value; NPV, negative predictive value; LR, likelihood ratio; IF, immunofluorescence.

#### 3.1.7. Criteria for Differentiating Class V LN and Non-Lupus Membranous Nephropathy—HCFMUSP

The factors correlated with lupus membranous nephropathy in the univariate analysis are presented in Table 5. Mesangial proliferation showed 68% sensitivity and 60% specificity, whereas mesangial deposits showed 76% sensitivity and 58% specificity. The immunofluorescence elements correlating with LN were IgA, IgM, and C1q, with 76% sensitivity. IgA showed the highest specificity (82%). C1q positivity ≥1+ increased specificity to 82%. Full-house staining showed 22% sensitivity and 93% specificity, and positivity for four factors (IgG, C3, and C1q with IgA or IgM) increased the sensitivity to 57%. The best performance was found when considering the positivity for four elements indiscriminately, with 61% sensitivity and 91% specificity. The following three factors remained in the multivariate analysis: mesangial deposits, C1q positivity, and four or more positive elements in immunofluorescence. The best performance occurred with a combination of two or more factors, with 76% sensitivity and 89% specificity (Appendix A). The area under the ROC curve was 85% (Appendix A).

### 3.2. HAN Population

#### 3.2.1. Histopathological Features Diagnostic Performance—HAN

The general characteristics and distribution of histopathological diagnoses of the HAN population are provided in the Appendix A. The LN group included mostly women (82.9%), with a median age of 29 years old, similarly to the HCFMUSP group. However, the HAN LN group included mostly black and mixed-race patients (65.2%), with a few white patients (8.5%) in contrast to the HCFMUSP LN group. Table 6 presents the sensitivity and specificity values of the histopathological elements for distinguishing LN from non-lupus glomerulopathy. Mesangial proliferation and endocapillary hypercellularity had sensitivity values of 69% and 74% and specificity values of 43% and 61%, respectively. Mesangial, subendothelial, and subepithelial deposits had sensitivities of 70%, 74%, and 49%, and specificities of 36%, 58%, and 55%, respectively. Extraglomerular deposits and combined subendothelial and subepithelial deposits had high specificity (90% and 100%, respectively) but low sensitivity. IgG staining showed a sensitivity of 96% and a specificity of 35%. C1q showed a sensitivity of 69% and a specificity of 77%. Dominant or co-dominant IgG improved its specificity to 48% but reduced its sensitivity to 85%. C1q ≥1+ positivity increased its specificity to 83% and reduced its sensitivity to 64%. Full-house staining showed 40% sensitivity and 97% specificity. IgG, C3, and C1q as requirements, and IgA or IgM as the fourth positive element had 58% sensitivity and 94% specificity. When considering positivity for four elements indiscriminately, the sensitivity was 64% and the specificity was 80%.

#### 3.2.2. LN Histopathological Criteria Combination—HAN

Subendothelial deposits, C1q positivity ≥1+, dominant or co-dominant IgG, and four or more positive elements in immunofluorescence remained independently related to LN and were selected for combined analysis. The sensitivity and specificity analyses for the combined elements are presented in Table 7. The best performance was achieved when two or more positive factors were combined, with 85% sensitivity and 65% specificity, and when three or more factors were combined, with 66% sensitivity and 92% specificity. Figure 2 shows an area under the ROC curve of 86%. These results are quite similar to those from the HCFMUSP, despite the ethnically diverse population, validating these histological findings.

**Table 6 diagnostics-14-02681-t006:** Diagnostic performance of histopathological features for lupus nephritis—HAN.

Feature	Sensitivity (IC 95%)	Specificity (IC 95%)	Accuracy (IC 95%)	PPV (IC 95%)	NPV (IC 95%)	Positive LR (IC 95%)	Negative LR (IC 95%)
Mesangial proliferation	0.69 (0.61, 0.76)	0.43 (0.32, 0.55)	0.60 (0.54, 0.67)	0.71 (0.64, 0.78)	0.40 (0.30, 0.51)	1.21 (0.97, 1.50)	0.73 (0.52, 1.02)
Endocapillary hypercellularity	0.74 (0.67, 0.81)	0.61 (0.49, 0.72)	0.70 (0.64, 0.76)	0.80 (0.72, 0.86)	0.53 (0.43, 0.64)	1.89 (1.42, 2.53)	0.42 (0.31, 0.58)
Mesangial deposits	0.70 (0.62, 0.77)	0.36 (0.25, 0.49)	0.59 (0.52, 0.66)	0.71 (0.63, 0.78)	0.35 (0.24, 0.47)	1.09 (0.89, 1.34)	0.84 (0.56, 1.24)
Subendothelial deposits	0.74 (0.66, 0.80)	0.58 (0.45, 0.70)	0.69 (0.62, 0.75)	0.80 (0.72, 0.86)	0.49 (0.38, 0.61)	1.75 (1.31, 2.35)	0.46 (0.33, 0.64)
Subepithelial deposits	0.49 (0.41, 0.57)	0.55 (0.43, 0.67)	0.51 (0.44, 0.58)	0.71 (0.61, 0.79)	0.33 (0.25, 0.42)	1.10 (0.82, 1.48)	0.92 (0.71, 1.19)
Combined subendothelial and subepithelial deposits	0.24 (0.18, 0.31)	0.90 (0.80, 0.96)	0.44 (0.37, 0.51)	0.84 (0.71, 0.94)	0.34 (0.27, 0.41)	2.37 (1.11, 5.04)	0.85 (0.75, 0.95)
Extraglomerular deposits	0.33 (0.26, 0.41)	1.00 (0.95, 1.00)	0.54 (0.48, 0.61)	1.00 (0.93, 1.00)	0.41 (0.34, 0.49)	Inf (NaN, Inf)	0.67 (0.60, 0.75)
Positive IgA	0.63 (0.55, 0.71)	0.55 (0.42, 0.67)	0.60 (0.53, 0.67)	0.75 (0.66, 0.82)	0.41 (0.31, 0.52)	1.39 (1.04, 1.86)	0.68 (0.50, 0.92)
Positive IgG	0.96 (0.91, 0.98)	0.35 (0.24, 0.48)	0.76 (0.70, 0.82)	0.76 (0.69, 0.82)	0.79 (0.60, 0.92)	1.47 (1.23, 1.76)	0.12 (0.05, 0.29)
Positive IgM	0.65 (0.57, 0.73)	0.44 (0.32, 0.57)	0.59 (0.51, 0.65)	0.71 (0.62, 0.79)	0.38 (0.27, 0.49)	1.17 (0.91, 1.49)	0.79 (0.55, 1.12)
Positive C3	0.84 (0.77, 0.90)	0.32 (0.21, 0.45)	0.68 (0.61, 0.74)	0.73 (0.66, 0.80)	0.49 (0.33, 0.65)	1.25 (1.04, 1.50)	0.48 (0.29, 0.81)
Positive C1q	0.69 (0.61, 0.77)	0.77 (0.65, 0.87)	0.72 (0.65, 0.78)	0.87 (0.79, 0.93)	0.54 (0.43, 0.64)	3.06 (1.93, 4.83)	0.40 (0.30, 0.52)
C1q ≥1+	0.64 (0.55, 0.72)	0.83 (0.72, 0.91)	0.70 (0.63, 0.76)	0.89 (0.82, 0.95)	0.51 (0.42, 0.61)	3.83 (2.20, 6.67)	0.43 (0.34, 0.55)
Dominant IgG	0.85 (0.78, 0.90)	0.48 (0.36, 0.61)	0.73 (0.66, 0.79)	0.77 (0.70, 0.84)	0.60 (0.46, 0.74)	1.65 (1.29, 2.10)	0.31 (0.20, 0.50)
Full-house staining	0.40 (0.32, 0.49)	0.97 (0.89, 1.00)	0.58 (0.51, 0.65)	0.97 (0.88, 1.00)	0.43 (0.35, 0.52)	13.20 (3.32, 52.46)	0.62 (0.54, 0.71)
IgG/C3/C1q plus IgA or IgM	0.58 (0.49, 0.66)	0.95 (0.87, 0.99)	0.70 (0.63, 0.76)	0.96 (0.90, 0.99)	0.51 (0.42, 0.60)	12.47 (4.09, 38.00)	0.44 (0.36, 0.54)
≥4 elements staining in IF	0.64 (0.55, 0.72)	0.80 (0.68, 0.89)	0.69 (0.62, 0.75)	0.87 (0.79, 0.93)	0.51 (0.41, 0.61)	3.20 (1.94, 5.29)	0.45 (0.35, 0.58)

HAN, Hospital Ana Nery; PPV, positive predictive value; NPV, negative predictive value; LR, likelihood ratio; IF, immunofluorescence.

## 4. Discussion

We described the histopathological findings of patients with LN from two nephrology referral centers and compared them with those of patients with other glomerular diseases that can mimic LN. The data were presented separately because of differences in clinical profiles, especially race composition, and the independent histopathological analysis between the centers. The HAN population served as a validation cohort for data from the HCFMUSP, with a socially and ethnically diverse sample. São Paulo ranks first in per capita income among Brazilian states, with 57.8% of its population declaring themselves as white in the last census. Bahia ranks 23rd out of 26 states in terms of income, with 79.7% of its population declaring themselves as black [20]. Because of this ethnically diverse population, the comparison between the two LN groups also reflects race influence on the development of LN.

Various histopathological features are associated with LN. At the HCFMUSP, features showing better diagnostic performance included mesangial proliferation, subendothelial deposits, C1q staining ≥1+, dominant IgG, and four or more positive elements in immunofluorescence. Individually, they exhibited sensitivity values between 74% and 92%, and specificity values between 36% and 89% for LN. The area under the curve (AUC) was 94% when combined and plotted on a ROC curve. Similar analyses were conducted for HAN, with findings largely consistent with those from HCFMUSP, except for mesangial proliferation. While mesangial proliferation showed a near-significant association in univariate analysis (*p* = 0.07), it did not remain significant in the multivariate analysis. The small sample size of the HAN population may have influenced the number of elements retained in the combined analysis. Despite these differences, including a diverse population, the results were similar between the two groups, strengthening the purpose of validation. Therefore, the utilization of those five histological features with the best diagnostic performance, as described above and subsequently validated in the HAN population, may constitute a valuable tool for LN diagnosis, especially in the absence of other SLE clinical and laboratory manifestations.

Few studies have defined LN based on histological findings. Kudose et al. identified intense C1q staining, full-house staining, tubuloreticular inclusions, extraglomerular deposits, and combined subepithelial and subendothelial deposits as significant features of LN. Their specificities ranged from 80% to 96%, sensitivities from 68% to 80%, and the ROC curve to differentiate LN from other diagnoses had an AUC of 96% [18]. Albeit with differences, the following two criteria described by Kudose et al. are similar to our study: intense C1q staining, which is a variant of the C1q staining ≥1+ in this cohort; and full-house staining, which was modified in our analysis to four or more positive elements in immunofluorescence. Both changes should be regarded as refinements that improve their performance as diagnostic criteria. Extraglomerular deposits and combined subepithelial and subendothelial deposits were analyzed, showing high specificities (97% and 98% for HCFMUSP and 100% and 90% for HAN). However, their low sensitivities (16% and 30% for HCFMUSP and 33% and 24% for HAN) limit their use as valid histological criteria. Tubuloreticular inclusions were not evaluated in this cohort because of the lack of routine electron microscopy.

In addition, full-house staining is highly suggestive of LN, even though it is considered a lupus-like synonyms by many authors [21]. We found a low sensitivity for this feature in both populations (37% for HCFMUSP and 40% for HAN). However, in the study by Kudose et al., the sensitivity was 71%. On the other hand, we found 96% specificity for HCFMUSP and 97% specificity for HAN, which is similar to that previously described (90% in Kudose et al.) [18]. In our study, selecting positivity for four or more elements in the final model provided a better balance between sensitivity and specificity, with comparable accuracy. The full-house pattern’s association with LN has been reinforced by various authors over the decades, but it remains essential to exclude other primary and secondary causes before establishing a diagnosis of idiopathic full-house nephropathy [22,23,24].

In the absence of a published consensus, the histological categorization of renal biopsy as LN by a nephropathologist depends on the combination of clinical history, laboratory evaluation, and histological features. Ahmed et al. have used a combination of the following criteria to classify a biopsy as lupus-like: (1) full-house staining in immunofluorescence with at least 1+ intensity, (2) tubuloreticular inclusions on electron microscopy, and (3) extraglomerular immune deposits with IgG and/or C1q [19]. Again, we found the best accuracy with four or more positive elements in immunofluorescence for LN diagnosis and could not analyze the other two factors because of the lack of EM.

Lupus membranous nephropathy (class V) differs clinically and histologically from the proliferative forms (classes III and IV) [10]. Those histological features which were found accurate for LN did not exhibit the same reliability in membranous LN. We analyzed the histopathological factors that could aid in distinguishing class V LN from non-lupus membranous nephropathy. Our analysis identified a combination of the following three factors: mesangial deposits, simple C1q staining, and four or more positive elements in immunofluorescence, with an area under the ROC curve of 85%. Mesangial proliferation is also associated with lupus membranous nephropathy, highlighting the significance of mesangial changes as predictors of SLE. Utilizing the same elements to differentiate between LN and other glomerular diseases, Kudose et al. achieved an AUC of 98% to distinguish anti-phospholipase A2 receptor (anti-PLA2R) positive membranous glomerulopathy from class V LN [18]. In Jennette’s study, significant histopathological findings in lupus membranous biopsies included mesangial dense deposits, small subendothelial dense deposits, tubuloreticular inclusions, and deposits in tubular basement membranes. In immunofluorescence, only C1q positivity was more prevalent in membranous LN, with a stronger correlation for intense staining (≥2+). Full-house staining was not assessed [17]. Thus, those three factors described in our study may help in distinguishing between class V LN and primary membranous nephropathy, which again may help in establishing membranous LN diagnosis.

This study has several limitations. First, due to its retrospective nature, which relied on medical records, there are some inherent limitations to this kind of study. To mitigate potential biases, the lead researcher was the only one responsible for data collection. Second, the histopathological findings were evaluated by a single nephropathologist at each center and in an unblinded manner. However, the involvement of experienced pathologists in reputable centers enhances the reliability of anatomical pathology reports. The lack of electron microscopy is another limitation because it prevented the evaluation of certain histological features, such as tubuloreticular inclusions, and reduced the sensitivity of other findings, such as extraglomerular deposits and combined subepithelial/subendothelial deposits. However, routine electron microscopy is not commonly performed in renal pathology laboratories in our country, particularly when LN is clinically suspected. Additionally, owing to the convenience of sampling, the LN group was larger than the control group, despite the latter including multiple renal pathologies. Renal diseases such as minimal change disease and focal segmental glomerulosclerosis, which contributed significantly to biopsies at the research sites, were excluded because of their distinct clinical presentation and diverse histology compared to LN. Finally, separate analyses may have minimized the power to detect statistically significant associations, particularly in the multivariate analyses. Nevertheless, even with an ethnically diverse population and an independent histological analysis, the main histological features with good accuracy for LN were similar at both sites.

Advances in renal histopathology incorporating new techniques, such as proteomic analysis, promise to enhance the accuracy of renal biopsy in identifying disease etiology. Recently, new serological and immunofluorescence markers have emerged to distinguish primary and secondary membranous nephropathies [25]. Such important biomarkers include anti-PLA2R [26] and anti-THSD7A antibodies [27], which are specific to the primary form, as well as exostosin (EXT-1 and EXT-2), as markers of membranous nephropathy secondary to autoimmune diseases, particularly LN [28]. While similar discoveries are not yet available for proliferative forms of LN, these advances complement findings from optical, electron, and conventional immunofluorescence microscopy, broadening diagnostic possibilities.

## 5. Conclusions

In conclusion, the combination of mesangial proliferation, subendothelial deposits, C1q staining ≥1+, dominant IgG, and four or more positive elements in immunofluorescence offers good accuracy in defining renal involvement in SLE in clinical practice. These findings were validated in an ethnically diverse population and may also hold great significance for scientific research and diagnostic consensus, contributing to a more precise and accurate LN diagnosis. This is particularly relevant following the assumption of renal biopsy-confirmed LN as a definitive criterion for SLE diagnosis.

## Figures and Tables

**Figure 1 diagnostics-14-02681-f001:**
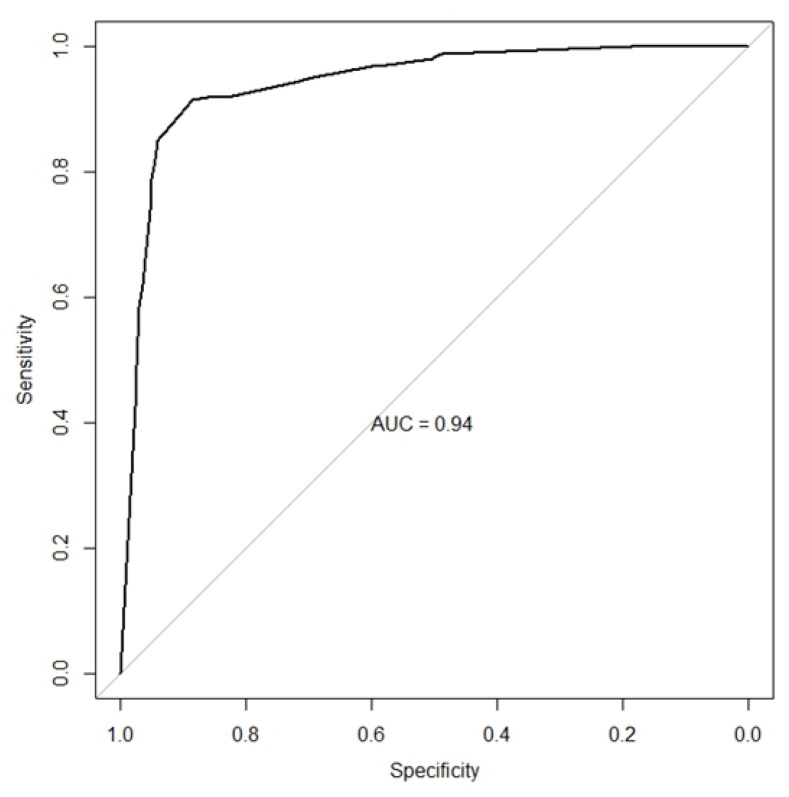
ROC curve to distinguish lupus nephritis and non-lupus glomerulopathies—HCFMUSP. The diagonal line defines an area under the curve of 50%.

**Figure 2 diagnostics-14-02681-f002:**
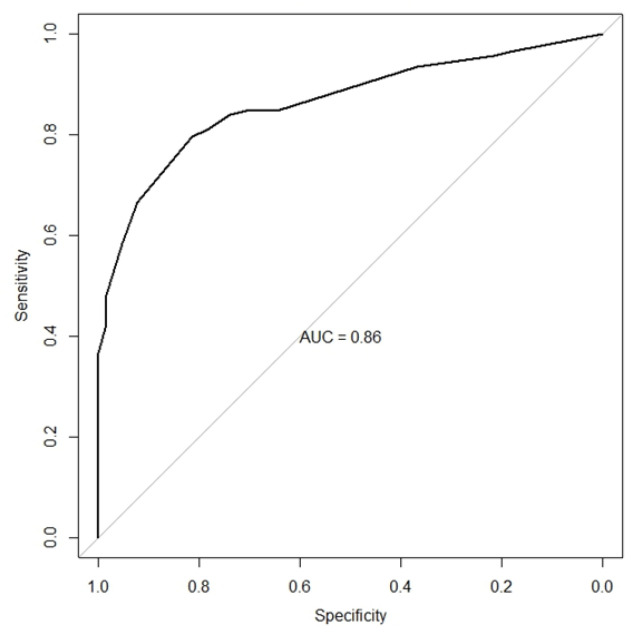
ROC curve to distinguish lupus nephritis and non-lupus glomerulopathies—HAN. The diagonal line defines an area under the curve of 50%.

**Table 1 diagnostics-14-02681-t001:** Demographic and clinical characteristics—HCFMUSP patients.

	Lupus Nephritis (*n* = 269)	Control (*n* = 219)	
** *Demographic data:* **			
Age	31.5 (25–40)	44 (32–58)	*p* < 0.001
Sex			*p* < 0.001
Male	41 (15.2)	121 (55.2)	
Female	228 (84.8)	98 (44.8)	
Race/Ethnicity			*p* < 0.001
Black	49 (18.2)	14 (6.4)	
Mixed	18 (6.7)	9 (4.1)	
White	183 (68.0)	168 (76.7)	
Yellow	0	5 (2.3)	
Unknown	19 (7.1)	23 (10.5)	
** *Clinical data:* **			
Creatinine (mg/dL):	1.00 (0.73–2.03)	1.96 (1.12–3.41)	*p* < 0.001
eGFR (CKD-EPI):	73 (29–110)	34 (17–71)	*p* < 0.001
Hemoglobin (g/dL):	10.9 (9.6–12.2)	11.1 (9.6–13.2)	*p* = 0.06
Albumin (g/dL):	3.0 (2.4–3.5)	3.2 (2.5–3.6)	*p* = 0.15
Cholesterol (mg/dL):	218 (173–274)	200 (164–252)	*p* = 0.06
Proteinuria (g/d or g/g):	2.010 (0.900–4.093)	2.300 (1.200–4.453)	*p* = 0.29
Hematuria (%):	186 (72.7)	144 (75.4)	*p* = 0.51
Positive ANA (%):	214 (98.2)	57 (32.7)	*p* < 0.001
Renal syndrome (%):			*p* < 0.001
Acute glomerulonephritis	35 (13.7)	47 (25.0)	
Nephrotic syndrome	15 (5.9)	25 (13.3)	
Nephritic/nephrotic syndrome	74 (29.0)	32 (17.0)	
Rapidly progressive GN	13 (5.1)	25 (13.3)	
Unknown etiology renal injury	15 (5.9)	17 (9.0)	
Asymptomatic urinary abnormalities	103 (40.4)	42 (22.3)	

All data are displayed as median (interquartile range) or *n*(%). HCFMUSP, Hospital das Clínicas of the University of São Paulo Medical School; eGFR, estimated glomerular filtration rate; CKD-EPI, Chronic Kidney Disease Epidemiology Collaboration; ANA, anti-nuclear antibody; GN, glomerulonephritis.

**Table 2 diagnostics-14-02681-t002:** Characterization of immune deposits in optic and immunofluorescence microscopy—HCFMUSP.

Immune Deposit Sites	Lupus Nephritis (n = 269)	Control (n = 219)	IgA Nephropathy (n = 88)	MN (n = 47)	MPGN (n = 31)	Pauci-Immune GN (n = 32)	Proliferative GN (n = 21)
Mesangial	220 (87.3)	165 (77.8)	88 (100)	19 (42.2)	25 (86.2)	17 (53.1)	16 (88.9)
Subendothelial	194 (73.8)	52 (24.2)	10 (11.4)	2 (4.3)	29 (93.5)	1 (3.1)	10 (55.6)
Subepithelial	130 (49.2)	51 (23.5)	0	47 (100)	1 (3.3)	0	3 (15.0)
Subendothelial and subepithelial	78 (29.8)	4 (1.8)	0	2 (4.3)	1 (3.3)	0	1 (5.0)
Tubular	36 (14.5)	3 (1.4)	0	0	1 (3.6)	0	2 (11.1)
Vascular	7 (2.8)	3 (1.4)	1 (1.1)	0	0	1 (3.1)	1 (5.6)
Extraglomerular	39 (15.7)	6 (2.8)	1 (1.1)	0	1 (3.5)	1 (3.1)	3 (16.7)
**Immunofluorescence positivity**							
IgA	141 (56.8)	102 (49.0)	88 (100)	8 (17.8)	4 (14.8)	0	2 (12.5)
IgG	242 (97.6)	83 (40.3)	11 (12.8)	45 (100)	14 (51.8)	4 (12.5)	9 (56.2)
IgM	172 (69.6)	89 (43.2)	36 (41.9)	15 (33.3)	23 (85.2)	10 (32.3)	5 (29.4)
Kappa	148 (59.9)	66 (32.0)	17 (19.8)	36 (80.0)	12 (44.4)	0	1 (5.9)
Lambda	222 (89.9)	133 (64.9)	73 (85.9)	41 (93.2)	15 (55.6)	0	4 (23.5)
C3	227 (91.5)	157 (75.5)	74 (85.1)	27 (60.0)	22 (84.6)	19 (59.4)	14 (82.3)
C1q	215 (86.7)	38 (18.3)	8 (9.3)	10 (22.2)	15 (55.6)	2 (6.2)	3 (17.6)
Fibrinogen	27 (10.8)	11 (5.3)	5 (5.7)	0	0	4 (12.9)	2 (11.8)
**Other**							
Full-house staining	91 (36.8)	8 (3.8)	2 (2.3)	3 (6.7)	2 (7.4)	0	1 (5.9)
Dominant IgG	227 (91.5)	66 (32.5)	0	45 (100)	11 (40.7)	3 (9.7)	7 (43.7)
C1q ≥1+	197 (79.4)	23 (11.1)	2 (2.3)	8 (17.8)	11 (40.7)	1 (3.1)	1 (5.9)
IgG/C3/C1q plus IgA or IgM	176 (71.3)	13 (6.2)	2 (2.3)	3 (6.7)	7 (25.9)	0	1 (5.9)
≥4 elements staining in IF	187 (75.7)	22 (10.7)	10 (11.6)	4 (8.9)	7 (25.9)	0	1 (6.2)

Note: Data are presented as n (%). Missing data were excluded from the analysis. HCFMUSP, Hospital das Clínicas of the University of São Paulo Medical School; MN, membranous nephropathy; MPGN, membranoproliferative glomerulonephritis; IF, immunofluorescence.

**Table 4 diagnostics-14-02681-t004:** Combined diagnostic performance of histopathological features for lupus nephritis–HCFMUSP.

No. of Features *	Sensitivity (IC 95%)	Specificity (IC 95%)	Accuracy (IC 95%)	PPV (IC 95%)	NPV (IC 95%)	Positive LR (IC 95%)	Negative LR (IC 95%)
One or more	1.00 (0.99, 1.00)	0.19 (0.14, 0.25)	0.63 (0.59, 0.68)	0.60 (0.55, 0.65)	1.00 (0.91, 1.00)	1.23 (1.15, 1.31)	0.00 (0.00, NA)
Two or more	0.96 (0.93, 0.98)	0.61 (0.54, 0.67)	0.80 (0.76, 0.84)	0.75 (0.70, 0.79)	0.93 (0.87, 0.97)	2.44 (2.06, 2.90)	0.06 (0.03, 0.12)
Three or more	0.91 (0.87, 0.94)	0.87 (0.81, 0.91)	0.89 (0.86, 0.92)	0.89 (0.85, 0.93)	0.89 (0.84, 0.93)	6.84 (4.81, 9.74)	0.10 (0.07, 0.15)
Four or more	0.74 (0.68, 0.80)	0.96 (0.92, 0.98)	0.84 (0.80, 0.87)	0.95 (0.91, 0.98)	0.75 (0.70, 0.81)	16.76 (8.81, 31.87)	0.27 (0.22, 0.33)
Five	0.46 (0.40, 0.53)	0.98 (0.94, 0.99)	0.69 (0.65, 0.74)	0.96 (0.90, 0.99)	0.60 (0.54, 0.65)	18.73 (7.80, 44.97)	0.55 (0.49, 0.62)

* Mesangial proliferation, subendothelial deposits, C1q ≥1+, dominant IgG and 4 or more positive elements in immunofluorescence. HCFMUSP, Hospital das Clinicas of the University of São Paulo Medical School; PPV, positive predictive value; NPV, negative predictive value; LR, likelihood ratio.

**Table 5 diagnostics-14-02681-t005:** Diagnostic performance of histopathological features in distinguishing class V lupus nephritis and non-lupus membranous nephropathy—HCFMUSP.

Feature	Sensitivity (IC 95%)	Specificity (IC 95%)	Accuracy (IC 95%)	PPV (IC 95%)	NPV (IC 95%)	Positive LR (IC 95%)	Negative LR (IC 95%)
Mesangial proliferation	0.68 (0.52, 0.81)	0.60 (0.44, 0.74)	0.64 (0.53, 0.74)	0.61 (0.46, 0.75)	0.67 (0.50, 0.80)	1.69 (1.13, 2.52)	0.53 (0.33, 0.87)
Mesangial deposits	0.76 (0.61, 0.87)	0.58 (0.42, 0.72)	0.67 (0.56, 0.77)	0.65 (0.51, 0.77)	0.70 (0.53, 0.84)	1.80 (1.23, 2.63)	0.41 (0.23, 0.73)
Positive IgA	0.43 (0.29, 0.59)	0.82 (0.68, 0.92)	0.63 (0.52, 0.73)	0.71 (0.51, 0.87)	0.59 (0.46, 0.71)	2.45 (1.20, 4.97)	0.69 (0.52, 0.92)
Positive IgM	0.70 (0.54, 0.82)	0.67 (0.51, 0.80)	0.68 (0.58, 0.78)	0.68 (0.53, 0.81)	0.68 (0.52, 0.81)	2.09 (1.32, 3.29)	0.46 (0.28, 0.74)
Positive C1q	0.76 (0.61, 0.87)	0.78 (0.63, 0.89)	0.77 (0.67, 0.85)	0.78 (0.63, 0.89)	0.76 (0.61, 0.87)	3.42 (1.94, 6.06)	0.31 (0.18, 0.53)
C1q ≥1+	0.65 (0.50, 0.79)	0.82 (0.68, 0.92)	0.74 (0.63, 0.82)	0.79 (0.63, 0.90)	0.70 (0.56, 0.82)	3.67 (1.89, 7.12)	0.42 (0.28, 0.64)
Full-house staining	0.22 (0.11, 0.36)	0.93 (0.82, 0.99)	0.57 (0.46, 0.67)	0.77 (0.46, 0.95)	0.54 (0.42, 0.65)	3.26 (0.96, 11.08)	0.84 (0.71, 1.00)
IgG/C3/C1q plus IgA or IgM	0.57 (0.41, 0.71)	0.93 (0.82, 0.99)	0.75 (0.65, 0.83)	0.90 (0.73, 0.98)	0.68 (0.55, 0.79)	8.48 (2.76, 26.04)	0.47 (0.33, 0.65)
≥4 elements staining in IF	0.61 (0.45, 0.75)	0.91 (0.79, 0.98)	0.76 (0.66, 0.84)	0.88 (0.71, 0.96)	0.69 (0.56, 0.81)	6.85 (2.61, 17.95)	0.43 (0.30, 0.62)

HCFMUSP, Hospital das Clinicas of the University of São Paulo Medical School; PPV, positive predictive value; NPV, negative predictive value; LR, likelihood ratio; IF, immunofluorescence.

**Table 7 diagnostics-14-02681-t007:** Combined diagnostic performance of histopathological features for lupus nephritis—HAN.

No. of Features *	Sensitivity (IC 95%)	Specificity (IC 95%)	Accuracy (IC 95%)	PPV (IC 95%)	NPV (IC 95%)	Positive LR (IC 95%)	Negative LR (IC 95%)
One or more	0.96 (0.92, 0.99)	0.18 (0.10, 0.30)	0.71 (0.65, 0.77)	0.71 (0.64, 0.78)	0.71 (0.44, 0.90)	1.18 (1.05, 1.33)	0.20 (0.07, 0.54)
Two or more	0.85 (0.78, 0.90)	0.65 (0.52, 0.76)	0.78 (0.72, 0.84)	0.83 (0.76, 0.89)	0.67 (0.54, 0.78)	2.39 (1.71, 3.35)	0.24 (0.15, 0.37)
Three or more	0.66 (0.58, 0.74)	0.92 (0.83, 0.97)	0.75 (0.68, 0.81)	0.95 (0.88, 0.98)	0.57 (0.47, 0.67)	8.64 (3.69, 20.21)	0.36 (0.28, 0.47)
Four	0.36 (0.28, 0.45)	1.00 (0.94, 1.00)	0.57 (0.50, 0.64)	1.00 (0.93, 1.00)	0.43 (0.35, 0.51)	Inf (NaN, Inf)	0.64 (0.56, 0.72)

* Subendothelial deposits, C1q ≥1+, dominant IgG and 4 or more positive elements in immunofluorescence. HAN, Hospital Ana Nery; PPV, positive predictive value; NPV, negative predictive value; LR, likelihood ratio; IF, immunofluorescence.

## Data Availability

The original contributions presented in the study are included in the Article/Appendix A, and further enquiries can be directed to the corresponding author (epitacio.rafael@gmail.com).

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
