# Peer review of "Analysis of the Sensitivity and Specificity of Histopathological Findings for Diagnosing Lupus Nephritis"

_diagnostics, 2024, doi:10.3390/diagnostics14232681_

Round 1

Reviewer 1 Report

Comments and Suggestions for Authors

I appreciated the opportunity to review this paper that investigates the diagnostic accuracy of histopathological findings in lupus nephritis. 

I just have some suggestions that I think can help improve the paper:

-I the introduction section, it would be useful to provide a brief summary of existing challenges in the histopathological diagnosis of LN

- in the methods, please explain in more detail how differences in diagnostic practices across the two centers (for instance the use of different staining methods) were managed to not alter the results of the study. Moreover, highlight how missing or incomplete biopsy data were handled in the analysis.

- Do you feel that the absence of electron microscopy may have impacted the ability to evaluate certain features? please explain in the manuscript this aspect

Author Response

  • Introduction section, it would be useful to provide a brief summary of existing challenges in the histopathological diagnosis of LN –

We have included a statement about the current challenges in the histological diagnosis of LN and the uncertainty about the specificity and accuracy of the main histological findings for the diagnosis of LN.

  • In the methods, please explain in more detail how differences in diagnostics practices across the two centers were managed to not alter the results of the study. Moreover, highlight how missing or incomplete data were handled in the analysis –

We have added in methods that the only difference between centers was the use of different preservation solutions in some samples, which has not impacted the histological analysis. All the staining methods were the same in both centers.

All missing and incomplete data were excluded from the analysis. This statement was included in the statistical analysis section.

  • Do you feel that the absence of EM may have impacted the ability to evaluate certain features? Please explain in the manuscript this aspect –

The absence of EM precluded the analysis of some histopathological characteristics, such as tubuloreticular inclusions, and reduced the sensitivity of other findings, such as extraglomerular deposits and combined subepithelial/subendothelial deposits, which are better identified in EM. Tubuloreticular inclusions were not included in our study, and extraglomerular deposits, as well as combined subendothelial/subepithelial deposits, were not retained in the final model due to their very low sensitivity, despite exhibiting good specificity. We have acknowleged this limitation in the discussion.

Reviewer 2 Report

Comments and Suggestions for Authors

Dear Authors,

You have conducted a very extensive analysis of the material and presented it well. However, I would like to return to the objective of your study, which is to describe the histological findings in LN biopsies and compare them with those of other glomerular diseases presenting similar clinical features, specifically identifying the histopathological findings most specific to LN in an ethnically diverse population.

Where in the analysis and conclusions is the ethnically diverse population reflected? At the end of the article, line 455, you mention addressing the current conceptual gap. Could you please clarify what this conceptual gap is?

Additionally, if you claim novelty in your study, then specificity and accuracy should be presented in a separate algorithm or table. Otherwise, please convince me of the novelty of the study beyond a large, well-presented volume of data.

Finally, is it possible to provide some practical commentary for doctors based on your work?

Author Response

  • Where in the analysis and conclusions is the ethnically diverse population reflected? At the end of the article, line 455, you mention addressing the current conceptual gap. Could you please clarify what this conceptual gap is? –

Reviewer is absolutely correct regarding this ethnically diverse population issue. Indeed, it was not clear at all. We have specifically included the race in both HCFMUSP and HAN populations. They are quite different: HCFMUSP sample was constituted of white (68%) and black and mixed-race (24.9%), while the HAN population was constituted of white (8.5%) and black and mixed-race (65.2%). The HCFMUSP results were validated in the HAN sample, despite the ethnically diverse population. These figures were included in centers’ population demographics and these results were discussed and included in the conclusions. Furthermore, this analysis is original because it was performed in a large Latin-American population. We hope that now the manuscript is clear about this issue.

Regarding the current conceptual gap in conclusions, we have removed this statement and have clarified the actual meaning of it.

  • Additionally, if you claim novelty in your study, then specificity and accuracy should be presented in a separate algorithm or table. Otherwise, please convince me of the novelty of the study beyond a large, well-presented volume of data –

The novelty of this study is represented by the largest histological analysis for LN diagnosis in a Latin-American population. Thus, the results about specificity and accuracy are new and original from a large database, which were validated in a entire different population (HAN group). These aspects were better described throughout the text, including the abstract.

  • Finally, is it possible to provide some practical commentary for doctors based on your work.-

We have included practical commentaries regarding the study results in aiding tools for LN diagnostic, especially when other SLE clinical and laboratory manifestations are lacking.

Reviewer 3 Report

Comments and Suggestions for Authors

The study evaluates combinations of specific histological features in lupus nephritis (LN) compared to other glomerular diseases. Although individual features have been studied before, the paper assesses how these features work together to improve diagnostic accuracy. The research is well-designed, with a large sample size and robust statistical analysis. However, several methodological limitations and points of clarification should be addressed by the authors.

·       In the Introduction, the research gap is not clearly defined. Please indicate what is missing in existing knowledge or practice regarding histologic criteria for LN. In addition, a few sentences should be included on how histologic findings influence diagnostic and treatment decisions, particularly in cases where clinical and serologic criteria are inconclusive.

·       In the Methods, please provide the data on how the specific control diseases (e.g., membranous nephropathy, IgA nephropathy) were chosen and how they compare to LN in terms of clinical presentation. Provide a clarification on why diabetic nephropathy and FSGS were excluded. Were these diseases excluded because they are easier to distinguish clinically, or were it for other reasons?

·       The selection process for multivariate logistic regression models and the basis for combining certain histopathological elements are unclear. Please explain why certain features, despite low sensitivity, were retained in the analysis.

·       Please consider expanding on the potential integration of these histological criteria with clinical and serological markers. How might this combination improve diagnostic accuracy?

·       In the Discussion, provide more detail on the potential clinical implications of the identified histological criteria. How might these findings impact diagnostic procedures or treatment decisions in practice?

·       The manuscript contains some minor grammatical errors.

Author Response

  • In the Introduction, the research gap is not clearly defined. Please indicate what is missing in existing knowledge or practice regarding histologic criteria for LN. In addition, a few sentences should be included on how histologic findings influence diagnostic and treatment decisions, particularly in cases where clinical and serologic criteria are inconclusive.

We have included and modified the paragraphs regarding this research gap. We hope that the revised introduction has accomplished this task.

  • In the Methods, please provide the data on how the specific control diseases (e.g., membranous nephropathy, IgA nephropathy) were chosen and how they compare to LN in terms of clinical presentation. Provide a clarification on why diabetic nephropathy and FSGS were excluded. Were these diseases excluded because they are easier to distinguish clinically, or were it for other reasons?

We have clarified in methods that the control glomerulopathies were chosen because of similar histological alterations, therefore amenable for comparison. On the other hand, some glomerulopathies, such as diabetes and FSGS were excluded because they have a distinct clinical and histological manifestation from LN.

  • The selection process for multivariate logistic regression models and the basis for combining certain histopathological elements are unclear. Please explain why certain features, despite low sensitivity, were retained in the analysis.

Multivariate analysis was performed to identify features independently correlated with LN, followed by multinomial logistic regression to determine the best combination of histopathological elements with high accuracy. These analyses were used to construct the receiver-operating characteristic (ROC) curves. We have included all histopathological features associated with lupus nephritis in the univariate analysis, even those with limited accuracy and sensitivity, and carried them into the multivariate analysis to identify the best combination of features that remained independently associated with lupus nephritis.

  • Please consider expanding on the potential integration of these histological criteria with clinical and serological markers. How might this combination improve diagnostic accuracy?

As explained in the manuscript, SLE diagnosis can be performed using a table with different points for each SLE clinical or laboratory manifestation. A sum of 10 or greater confirms the SLE diagnosis. Renal histological manifestation, especially class III and IV, provides 10 points, which is enough to establish SLE diagnosis. Therefore, biopsy-proven LN is important for the disease diagnosis, especially when other SLE manifestations are absent.

  • In the Discussion, provide more detail on the potential clinical implications of the identified histological criteria. How might these findings impact diagnostic procedures or treatment decisions in practice?

We have added several statements and explanations about the relevance of the histological analysis for the LN diagnosis and the impact on disease diagnosis and treatment.

  • The manuscript contains some minor grammatical errors.

The manuscript was reviewed by an editing service company.